# Effects of Multi-Dike Protection Systems on Surface Water Quality in the Vietnamese Mekong Delta

**Huynh Vuong Thu Minh [1,2] Masaaki Kurasaki [1,2,\*], Tran Van Ty [3], Dat Quoc Tran [3,4], Kieu Ngoc Le [3,4], Ram Avtar [1,2], Md. Mostafizur Rahman [5] and Mitsuru Osaki [6]**

[1] Graduate School of Environmental Science, Hokkaido University, Sapporo 060-0810, Japan; hvtminh@ctu.edu.vn

[2] Faculty of Environmental Earth Science, Hokkaido University, Sapporo 060-0810, Japan; ram@ees.hokudai.ac.jp

[3] Can Tho University, Can Tho City 900000, Vietnam; tvty@ctu.edu.vn

[4] The University of Arkansas at Fayetteville, Fayetteville, AR 72701, USA; datquoct@uark.edu (D.Q.); knle@uark.edu (K.N.L.)

[5] Department of Environmental Sciences, Jahangirnagar University, Dhaka 1342, Bangladesh; rahmanmm@juniv.edu

[6] Research Faculty of Agriculture, Hokkaido University, Sapporo 060-0810, Japan; mosaki@chem.agr.hokudai.ac.jp

\* Correspondence: kura@ees.hokudai.ac.jp; Tel.: +81-11-706-2243

**Abstract:** The Vietnamese Mekong Delta (VMD) is one of the largest rice-growing areas in Vietnam, and exports a huge amount of rice products to destinations around the world. Multi-dike protection systems have been built to prevent flooding, and have supported agricultural intensification since the early 1990s. Semi-dike and full-dike systems have been used to grow double and triple rice, respectively. Only a small number of studies have been conducted to evaluate the water quality in the VMD. This study aimed to analyze the spatiotemporal variation of water quality inside the dike-protected area. Surface water samples were collected in the dry and wet seasons at 35 locations. We used multivariate statistical analyses to examine various water quality parameters. The mean concentrations of COD, $NH_4^+$, $NO_3^-$, $PO_4^{3-}$, EC, and turbidity were significantly higher in water samples inside the full-dike system than in water samples from outside the full-dike systems and inside the semi-dike systems in both seasons. High concentrations of $PO_4^{3-}$ were detected in most of the primary canals along which residential, tourist areas and local markets were settled. However, $NO_3^-$ was mainly found to be higher in secondary canals, where chemical fertilizers were used for rice intensification inside the dike system. Water control infrastructures are useful for preventing flood hazards. However, this has an adverse effect on maintaining water quality in the study area.

**Keywords:** multi-dike-protection systems; rice intensification; multivariate analysis; spatiotemporal variation of water quality

## 1. Introduction

Water quality has become a major concern in the environmental debate [1–4], and about 80% of the world's population currently faces the threat of water scarcity [2]. Complex and diffuse pollutants are transferred from multiple land use such as agricultural activity to surface water; thus, maintaining water quality is challenging [5–12] Agriculture is considered a primary source of livelihood for 40% of the population and feeds more than seven billion people in the world [13]. The

intensification of agriculture has been implemented to meet the rising food demand in conjunction with the limitation of land due to excessive urban development [11,14].

In Southeast Asia, the irrigated rice system area accounts for more than 50% of the total agricultural area in Indonesia, the Philippines, and Vietnam, and rice intensification often results in more than one crop per year and accounts for 79.4% of the total area under rice cultivation [15]. Also, it was found that the irrigated rice fields require more fertilizers compared to rain-fed rice cultivation [16]. In the VMD, agricultural intensification has been applied for nearly 40 years to adapt to economic development and population growth. In particular, the Renovation policies (a set of economic reforms aimed at moving the Vietnamese economy towards a market-based economy) issued by the Vietnamese government in 1986 have had a role in leading Vietnam to become one of the largest rice exporting countries in the world since the 2010s [17–21].

The VMD has experienced a significant hydrodynamic alteration, especially after the mighty floods in 2000 [22,23]. Many multi-dike protection, canals and sluice gates were built in the upper part of the VMD, such as in An Giang, Dong Thap, and Long An provinces, to prevent flood and manage irrigation and drainage. The VMD has therefore been transformed into a hydraulic landscape under human control since the 1990s [24–26]. In addition, Tran et al. [27] also found that the series of dams in the upstream of the Mekong River causes a change in water regime in the downstream areas. Furthermore, the use of pesticide and herbicide chemicals has swelled to wasteful levels in many parts of the region, where the local government is seeking solutions to reduce the use of the chemicals to improve water quality, especially in areas inside the dike systems. An Giang province, a prominent area in the VMD, with sufficient impact of farming and renovation policies, was chosen as the study area.

In An Giang, groundwater has become an alternative source of fresh water supply for domestic water use, while surface water has mainly been employed for raising fish and irrigation in recent years. The surface water quality often exceed the permissible limit of the Vietnamese standards for domestic water supplies, and it is recommended that surface water be treated before drinking and cooking [28]. Intensive rice cultivation has caused a changing ecosystem, decreasing water quality and potential toxicity, especially inside the dike system [29–33]. Variation in water quality depends on the location, time, season and existing pollution sources, and multivariate statistical techniques are able to assess and identify spatial and temporal variations in water quality. Hierarchical Cluster Analysis (HCA), Principal Component Analysis (PCA), Factor Analysis (FA) and Discriminant Analysis (DA) have been widely applied for characterizing and evaluating water quality worldwide [34–36]. Thus, we applied multivariate statistical analysis and water quality indexing to analyze spatial and temporal variations in surface water quality in canals inside and outside the multi-dike system and to identify the impacts of land use change on hotspots of water quality. This study aimed to assess the changes in surface water quality to find out the level of discriminant water quality parameters and their seasonal variation between the inside and the outside of dike system.. The findings of this study could provide a better understanding for improving the water quality situation in An Giang, and will be useful in assisting the An Giang government to take sustainable measures.

## 2. Materials and Methods

### 2.1. Site Description

The An Giang province (Figure 1) is located in the VMD, where the wet season occurs from May to November, and the dry season occurs from December to April. The average annual rainfall is 1400 mm, with 90% of the rainfall occurring during the monsoon season, from May to November. Water resources in the province come mostly from the upstream of the Mekong river, with rainfall runoff in An Giang having minimal effect on river flow [37]. The two main rivers of the Mekong River system, i.e., the Mekong and the Bassac rivers, along with 280 primary rivers and dense main canals, are responsible for the release of flood flow in An Giang, since the province is a flood-prone area.

Twenty-two high-frequency flood events happened between 1926 and 2006 [38], with about 65% of the high flood events occurring from August to October [39].

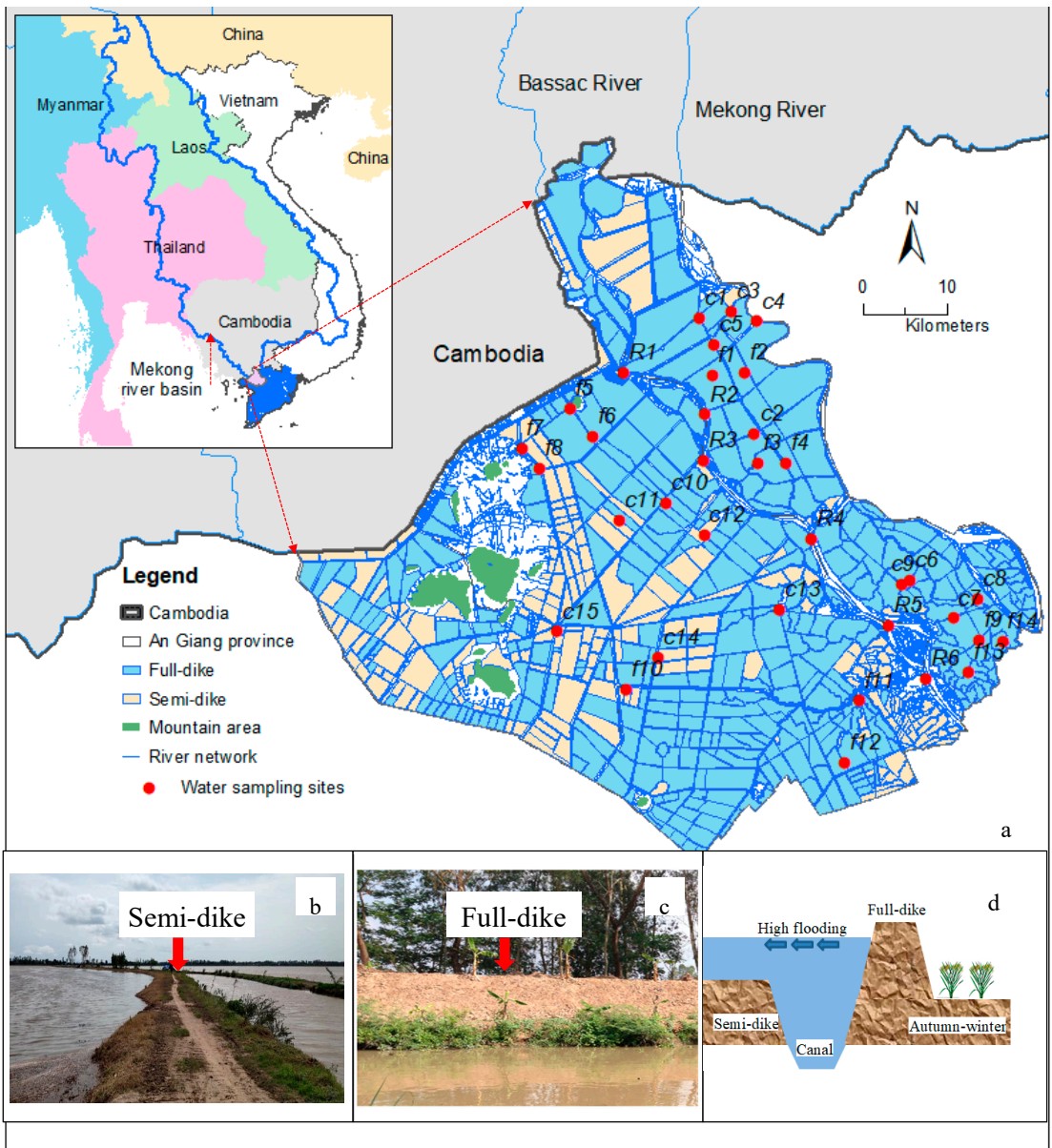

**Figure 1.** (**a**) Study area and surface water quality monitoring location in An Giang province, Vietnam. (**b**,**c**) Photos of a semi-dike and a full-dike (taken during field work in the Mekong Delta in 2017). (**d**) Sketch of a cross-section of a typical dike system. Note: R: sampling site in the main river, c: sampling site in the primary canal, f: sampling site in the secondary canal (field canal).

The multi-dike protection area accounted for over 80% of the total area of An Giang in 2017 (Figure 1). Farmers are able to practice triple-rice intensification in full-dike protection systems even during flooding season [40]. The full-dike system protects triple-rice crops with variation in elevation from 4 to 6 m above sea level [41,42]. Meanwhile, double-rice crops often come under semi-dike protection systems, with elevation ranging from 2 to 3 m above sea level. In this dike system, water flows into the fallow field periodically during monsoon season, during August and November [40]. Figure 2 shows the increased and decreased trend of the full-dike and semi-dike areas, respectively, during 1995 and 2017. The local farmers shifted to a triple-rice crop from a

double-rice crop, along with an increase in area covered by the full-dike system since 1995. Triple-rice crops have become predominant since the 2000s [41,43].

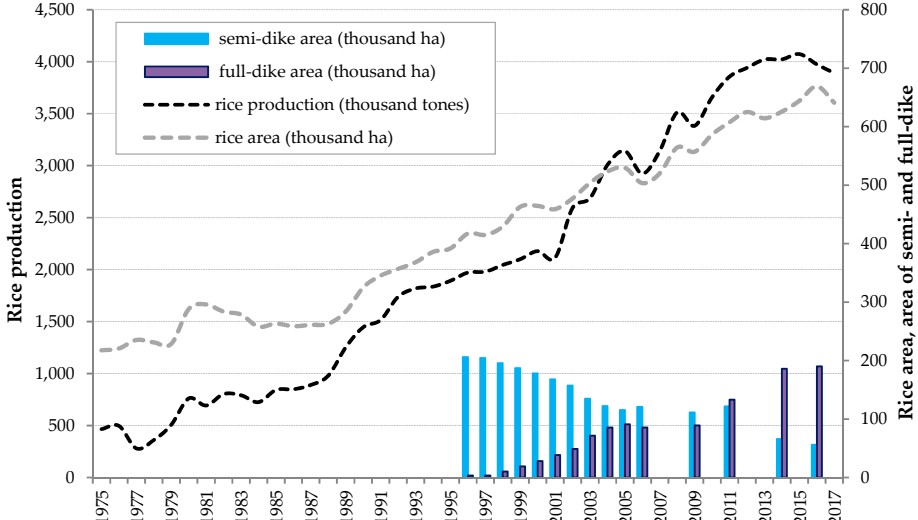

**Figure 2.** Rice area and production graph in multi-dike protection systems. Rice production was doubled from 2 million tons in 1997 to 4 million tons in 2017. Rice area and production data were collected from IRRI, spanning the 1975–2017 period; multi-dike area data were collected from Department of Agriculture and Rural Development (DARD) and Construction Department of An Giang during the period 1995–2016.

### 2.2. Water Quality Sampling and Analysis

Water quality samples were taken during the dry season (22–28 April 2017) and the wet season (6–13 October 2017). 35 surface water samples per season were taken at different rivers and canals inside and outside dike systems in An Giang. We conducted geotagged photography of the water sampling sites, which were then marked in the Global Positioning System (GPS). The stratified random sampling technique was used to select the sampling sites: sites R6 in the Bassac River (a branch of the Mekong River), site c15 in the primary canals, and f14 in the secondary canals (Figure 1). The physical parameters were measured in situ using a HORIBA multi-parameter meter (Kyoto, Japan) and a handheld meter (Oaklom; Tokyo, Japan). Chemical parameters such as phosphorus, nitrate, nitrite, ammonium, and chemical oxygen demand (COD) were measured using a pack test. Rice production data and dike system area in An Giang were collected from the Department of Agriculture and Rural Development (DARD) and Construction Department of An Giang, 1995–2017.

The multivariate approaches for the matrix for water quality parameters offer a better understanding of the characteristics of water quality [34,44–46]. Firstly, we assessed the temporal variation of water quality between the dry and wet seasons by using the Spearman correlation coefficient (Spearman R) and discriminant analysis (DA) techniques. Secondly, the cluster analysis (CA) and principal component analysis (PCA) were then applied to the spatial pattern assessment parameters for each season in order to detect the differences in water quality between the inside and outside of the dike. Finally, we calculated the water quality index based on clusters obtained in the CA step. The statistical software and data analysis add-on for Excel (XLSTAT) version 2018 and inverse distance weighting (IDW) interpolation were chosen to display the results, as this has been successfully applied in many previous studies [47–49].

The DA technique was used to evaluate changes in water quality linked with seasons. It can also be used to determine the most significant parameters, and can separate two or three clusters in the data sets [50–53]. In this study, we implemented three different modes, i.e., standard, forward stepwise and backward stepwise, which had been successfully applied in previous studies [54,55]. In

the forward stepwise mode, the variables are included step-by-step, beginning with the most significant improvement of fit until no changes are obtained. In the backward stepwise mode, the variables are removed step-by-step beginning with the least significant improvement of the fit until there are no significant changes, as expressed in Equation (1) [50,56].

$$f(G_i) = k_i + \sum_{i=1}^{n} w_{ij} p_{ij} \tag{1}$$

where $i$ is the number of clusters (G), $k_i$ is the constant inherent to each cluster, $n$ is the number of parameters, $w_j$ is the weight coefficient assigned by DA to a given selected parameter ($p_j$).

The F test of Wilks' lambda identifies the parameters that contribute significantly, i.e., a decrease in the independent variable's lambda value means an increase in the variable's contribution. Significantly contributing variables will be identified on this basis [56].

### 2.3. Spatial Pattern Water Quality Analysis

1. The CA tool, an unsupervised model, was applied to examine the spatial and temporal differences. This tool had been applied previously for water quality assessment [52,53,57–59]. In this study, CA was chosen to divide the data set into clusters within 35 water sampling sites. The most common approach starts at each site with the cluster that is most similar to a predetermined selection criterion. Then, the sites are joined together in a separate cluster until only one cluster remains [51–53,60–63]. For example, agglomerative hierarchical clustering with a bottom-up approach was applied, wherein each site starts in its own cluster, and then pairs of clusters are merged, moving up the hierarchy. Ward's method measures the distance between linked clusters, in which Dlink/Dmax represents the ratio of the linkage distances of the identified cluster to maximal linkage distance [50,53,62,64,65].

2. PCA was used to transform the original variables into new principal components, performed along the directions of maximum variance. Additionally, ways were identified of reducing the contribution of the less significant variables with minimal information loss. In this study, the principal component was applied to identify which factors were the most important parameters of water quality, as expressed by Equation (2) [59,66–73].

$$Z_{ij} = a_{1i} x_{1j} + a_{2i} x_{2j} + \cdots + a_{im} x_{nj} \tag{2}$$

where $Z$ is the component score, $a$ is the component loading, $x$ is the component number, $j$ is the sample number, and $m$ is the total number of variables.

3. FA was used to reduce the contribution of less significant variables in order to further simplify the data structure produced by PCA. Factor analysis is expressed by Equation (3).

$$Z_{ji} = a_{f1} f_{1i} + a_{f2} f_{2i} + \cdots + a_{fm} f_{mi} + e_{fi} \tag{3}$$

where $Z$ is the measured variable, $a$ is the factor loading, $f$ is the factor score, $e$ is the residual term accounting for errors or other sources of variation, $i$ is the sample number, and $m$ is the total of the factors.

### 2.4. Water Quality Assessment

The water quality index (WQI), which has been applied in many countries, as well as in the Mekong River basin, is one of the more prominent methods for representing the level of water quality, and combines several physico-chemical and biological parameters into a single number [35,36,74–80]. The standard of water quality for the whole lower Mekong River basin was evaluated in An Giang. However, there is no specific water quality standard or guideline that can be used as the criterion in the Lower Mekong Delta basin [74–76]. Thus, we used the water quality standard introduced by the Mekong river commission (MRC) in 2012 [77]. These six indicators of water quality, which concern aquatic life (pH, EC, ammonia ($NH_3^+$), nitrite and nitrate-nitrogen ($NO_{2,3}\_N$), total phosphorus (T-P), and DO), were calculated [76]. We used the water quality index for aquatic life protection (WQI_al) to evaluate surface water quality in An Giang, as expressed in Equation (4) [77].

$$WQI_{al} = \frac{\sum_{i=1}^{n} p_i}{M} \, x \, 10 \qquad (4)$$

where $M$ is the maximum possible score for the measured parameters; $n$ is the number of samples; $p_i$ is the points scored on the sample (when each of the parameters meets its threshold values in Table 1, its corresponding weighting factor is scored; otherwise, the score is zero). The thresholds for the six water quality parameters were used to estimate $WQI_{al}$. The rating scale for water quality assessment of aquatic life protection includes four-grade scales: 10–9 for high quality, 9.5–9 for good quality, 9–7 for moderate quality, and <7 for poor quality [75,76].

**Table 1.** Water quality parameters used for calculating the rating score of the water quality index for the protection of aquatic life.

| No. | Parameters | Threshold Values |
|-----|-----------|------------------|
| 1 | pH | 6–9 |
| 2 | EC ($Scm^{-1}$) | <150 |
| 3 | $NH_3^+$ ($mgL^{-1}$) | 0.1 |
| 4 | $NO_{2,3}$_N ($mg\ L^{-1}$) | 5 |
| 5 | T_P ($mg\ L^{-1}$) | 0.13 |
| 6 | DO ($mg\ L^{-1}$) | >5 |

Note: Total-$PO_4^{3-}$ ($mg\ L^{-1}$ as $PO_4^{3-}$) = $PO_4^{3-}$ × 0.3262, and $NH_3$ = $NH_4^+$ × 0.944, and $NO_2^-$_N = $NO_2^-$/3.28442, $NO_3$_N = $NO_3^-$/4.42664 [81,82].

## 3. Results

Tables 2 and 3 show the correlations among ten water quality parameters in the dry and wet seasons, respectively. The results indicated that water quality in the dry season was significantly different from its quality in the wet season. There were higher correlations among parameters in the dry season compared to those in the wet season. For example, in the dry season, turbidity had a statistically positive correlation with EC and $PO_4^{3-}$ concentrations ($p<0.05$), but no statistically significant correlation was found with other parameters in the wet season.

The pH value was significantly correlated with four parameters (i.e., EC, Turb, $NO_3^-$, and $PO_4^{3-}$) in the dry season, whereas it was only significantly correlated with EC in the wet season. Only EC had a significant correlation with more than one variable, i.e., COD, $NH_4^+$, and $PO_4^{3-}$ in the wet season. In contrast, we found that in the dry season, EC and pH were significantly correlated with three and four water quality parameters, respectively. Therefore, it can be expected that the water quality parameters varied widely in the two seasons.

**Table 2.** Correlation matrices in the dry season using Spearman rank order.

| Variables | DO | pH | EC | Turb | COD | $NH_4^+$ | $NO_2^-$ | $NO_3^-$ | $PO_4^{3-}$ | TC |
|-----------|------|--------|--------|--------|-------|-------|--------|--------|-------|----|
| DO | 1 | | | | | | | | | |
| pH | −0.001 | 1 | | | | | | | | |
| EC | −0.108 | −0.664 * | 1 | | | | | | | |
| Turb | −0.276 | −0.463 * | 0.657 * | 1 | | | | | | |
| COD | 0.076 | −0.068 | 0.310 | 0.394 | 1 | | | | | |
| $NH_4^+$ | −0.234 | −0.275 | 0.572 * | 0.495 * | 0.224 | 1 | | | | |
| $NO_2^-$ | 0.227 | 0.273 | 0.133 | 0.082 | 0.108 | 0.208 | 1 | | | |
| $NO_3^-$ | 0.139 | 0.505 * | −0.106 | −0.115 | 0.101 | 0.095 | 0.795 * | 1 | | |
| $PO_4^{3-}$ | −0.082 | −0.454 * | 0.504 * | 0.662 * | 0.396 | 0.298 | 0.054 | −0.030 | 1 | |
| TC | −0.004 | 0.112 | 0.177 | 0.013 | 0.123 | 0.101 | 0.109 | −0.012 | −0.225 | 1 |

Note: * indicates the correlation value is different from 0 at a significant level of 0.05.

**Table 3.** Correlation matrices in the wet season using Spearman rank order.

| Variables | DO | pH | EC | Turb | COD | $NH_4^+$ | $NO_2^-$ | $NO_3^-$ | $PO_4^{3-}$ | TC |
|---|---|---|---|---|---|---|---|---|---|---|
| **DO** | 1 | | | | | | | | | |
| **pH** | 0.059 | 1 | | | | | | | | |
| **EC** | −0.016 | −0.532 * | 1 | | | | | | | |
| **Turb** | −0.193 | 0.261 | −0.174 | 1 | | | | | | |
| **COD** | 0.055 | −0.387 | 0.486 * | −0.098 | 1 | | | | | |
| **$NH_4^+$** | −0.132 | −0.229 | 0.508 * | −0.176 | 0.380 | 1 | | | | |
| **$NO_2^-$** | −0.145 | −0.139 | 0.223 | 0.168 | 0.237 | 0.371 | 1 | | | |
| **$NO_3^-$** | −0.100 | −0.336 | 0.149 | 0.001 | 0.202 | 0.016 | 0.506 * | 1 | | |
| **$PO_4^{3-}$** | 0.039 | −0.162 | 0.484 * | 0.021 | 0.284 | 0.382 | −0.013 | −0.244 | 1 | |
| **TC** | −0.122 | −0.117 | 0.045 | −0.473 | 0.073 | 0.111 | 0.262 | 0.182 | 0.011 | 1 |

Note: * indicates the correlation value is different from 0 at a significant level of 0.05.

## 3.1. Temporal Variation of Water Quality Parameters

Table 4 shows the discriminant factions (DFS) on the three modes used in DA (please see Appendix Table A3 for the classification matrices (CMs)). In the standard mode, a DFS with six discriminant variables was obtained (COD, turbidity, EC, DO, $NO_3^-$, and $NH_4^+$), producing CMs that were 95.54% correct. Similarly, both backward stepwise and forward stepwise modes were 92.68% correct. Additionally, the test of Wilks Lambda in the standard mode provided a value of 0.301 ($p < 0.0001$). The null hypothesis states that the means of the vectors of the two groups (the dry and wet seasons) are equal. The alternative hypothesis, on the other hand, states that at least one of means of the vectors will be different from another. In this case, the computed $p$-value was lower than 0.01; the null hypothesis was rejected. Moreover, a $p$-value lower than 0.0001 indicates that the means of two groups were strongly different. Therefore, the concentrations of DO, EC, turbidity, COD, $NH_4^+$, and $NO_3^-$ were the most significant parameters in discriminating between the two seasons in the study area.

Figure 3 shows the concentrations of six parameters in the dry and wet seasons. Overall, the concentrations in both seasons almost failed to meet the Vietnamese water quality standard for domestic water supply presented in [83], and these findings are consistent with the findings of [28,33,79,84]. The concentrations of COD, Turb, and $NH_4^+$ in the wet season were higher in value and wider in range when compared with those in the dry season. Some high outliers of EC, COD, and $NH_4^+$ were found during the wet season, and most of these outliers were found at sites located in secondary canals inside full-dike systems, such as c7, f1, f3, f6, f9, and f14. We expect that the concentration of water quality parameters in semi-dike systems and outside dikes would be lower than that inside full-dike systems.

**Table 4.** Unidimensional lambda test of equality of water quality parameters.

| Variable | Standard Mode | Backward Stepwise | Forward Stepwise |
|---|---|---|---|
| DO | 0.808 *** | 0.808 *** | 0.808 *** |
| pH | 0.992 | | |
| EC | 0.812 *** | 0.812 *** | 0.812 *** |
| Turb | 0.790 *** | 0.790 *** | 0.790 *** |
| COD | 0.755 *** | 0.755 *** | 0.755 *** |
| $NH_4^+$ | 0.896 ** | 0.896 ** | 0.896 ** |
| $NO_2^-$ | 1.000 | | |
| $NO_3^-$ | 0.874 ** | 0.874 ** | 0.874 ** |
| $PO_4^{3-}$ | 0.975 | | |
| TC | 0.961 | | |
| Percent correct | 95.54% | 92.68% | 92.68% |

Note: Significance levels are denoted as follows: ** $p < 0.01$, *** $p < 0.001$. A Lambda value of 1 indicates that the means of water quality parameters in the dry season were not different from those

in the wet season. A Lambda value of 0 indicates that the means of water quality parameters in the dry season were totally different from those in the wet season.

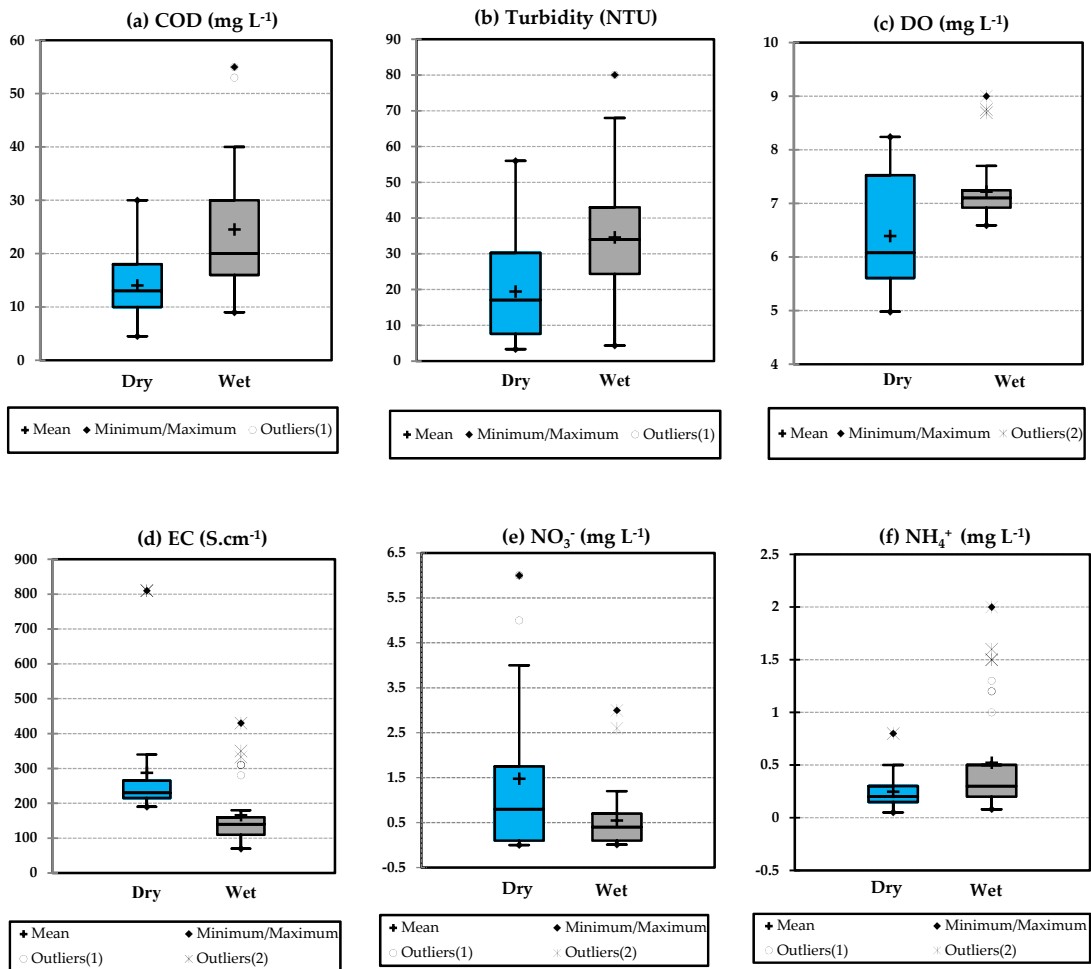

**Figure 3.** Summary data of discriminant parameters in the dry and the wet seasons: (**a**) COD, (**b**) turbidity, (**c**) DO, (**d**) EC, (**e**) $NO_3^-$, and (**f**) $NH_4^+$ in surface water quality. The outliers of $NH_4^+$ and EC in the wet season were found inside full-dike systems (c7, f1, f3, f6, f9, and f14). Note: R: river, c: primary canal, f: field canal; 1: inside full-dike, 2: inside semi-dike, 3: outside of dike.

*3.2. Spatial Variation of Water Quality Parameters*

Figure 4 shows the dendrogram clusters in the dry and wet seasons in which three statistically significant clusters with linkage distance (dissimilarity) represented at (Dlink/Dmax) × 100 < 60 (cluster 1, 2 and 3) and at (Dlink/Dmax) ×100 < 35 for sub-clusters (3a and 3b). On the other hand, the similarity distance among clusters indicates that the clusters still had similar characteristics with respect to water quality due to natural background source types. This finding was in agreement with findings from previous studies in the Asia region [54,55]. Figures 4 and 5 show a large discriminant between cluster 1 and the other clusters. The number of clusters was also decided based on the practicality of the results, such as the volume of available information, for the main river (R), primary canal (c), secondary canal (f), and full-dike (1), semi-dike (2), outside dike (3). The majority sites of cluster 1 were located inside the full-dike system in both seasons (Figure 4). In cluster 2, four out of nine sites were located in the full-dike system in the dry season, and seven out of nine sites were in the full-dike system during the wet season. Cluster 3a and 3b obtained mixed types such as

inside the full-, semi-, and outside of dike systems, except that cluster 3a in the dry season included the most sites that were located inside the full-dike system. We present some of the water quality characteristics by clusters, such as COD and $NH_4^+$ in the wet season, and $NO_3^-$ and $PO_4^{3-}$ in the dry season, in terms of average value and standard deviation. The average concentrations of COD were 40 (±13.37), 22 (±12), 26 (±7.5), and 15 (±3.3) in clusters 1, 2, 3a, and 3b, respectively, in the wet season. Furthermore, we found that the average values of $NH_4^+$ (1.23, ±0.5), $NO_3^-$ (0.58, ±0.65), and $PO_4^{3-}$ (0.7, ±0.51) in cluster 1 were higher than in other clusters.

Cluster 1 was obtained in secondary canals that were close to rice fields (eight sites in the dry season: f1, f4, f7, f8, f9, f14, c1, and c11; and six sites in the wet season: f1, f3, f6, f9, f14, and c1). Cluster 2, in the dry season, included a majority of sample sites located in the main river, as well as primary and secondary rivers. However, in the wet season, most samples sites (six out of nine) were located in the primary canals. Cluster 3a included sample sites located in secondary canals (five out of ten), such as f3, f5, f6, f10, f13, c4, c9, c10, R3, and R4, and primary canals inside semi-dike (f5, f8, f13, c6, c10, c12, c13, and c14) in the dry season. Cluster 3b included sites inside full-dike in the dry season and a mixture of types of sample site in the wet season.

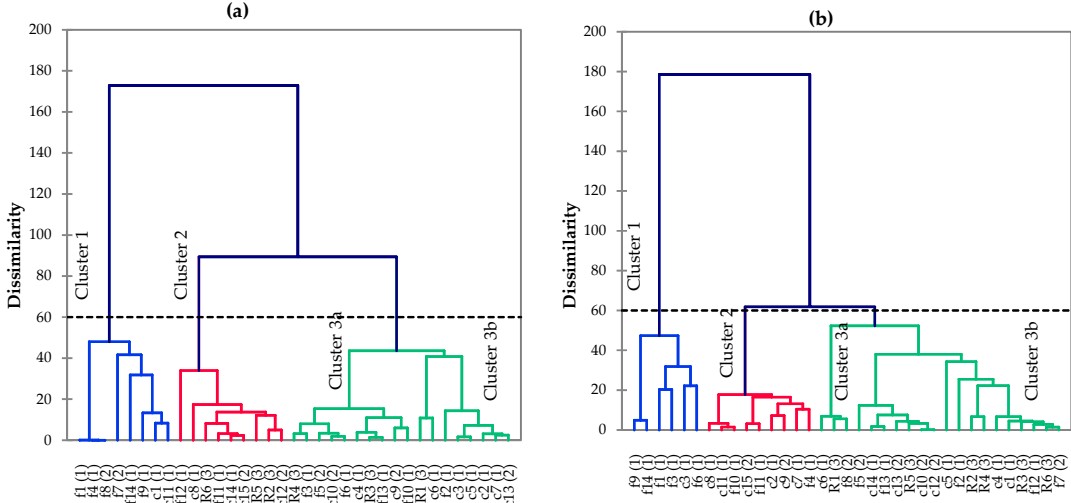

**Figure 4.** Dendrogram of clusters of sampling sites according to surface water qualities at (Dlink/Dmax) × 100 < 60 and (Dlink/Dmax) × 100 < 35 for sub-clusters (3a and 3b): (**a**) dry season and (**b**) wet season. In the dry season (cluster 1 obtained f1, f4, f7, f8, f9, f14, c1, and c11; cluster 2: f1, f12, c8, c12, c14, c15, R2, R5, and R6; cluster 3a: f3, f5, f6, f10, f13, c4, c9, c10, R3, and R4; cluster 3b: f2, c2, c3, c5, c6, c7, c13, and R1). In the wet season (cluster1: f1, f3, f6, f9, f14, and c1; cluster 2: f4, f10, f11, c2, c7, c8, c9, c11, and c15; cluster 3a: f5, f8, f13, c6, c10, c12, c13, c14, R1, and R5; cluster 3b: f2, f7, f12, c1, c4, c5, R2, R3, R4, and R6). Note: (R: river, c: primary canal, f: field canal; 1: inside full-dike, 2: inside semi-dike, 3: outside of dike).

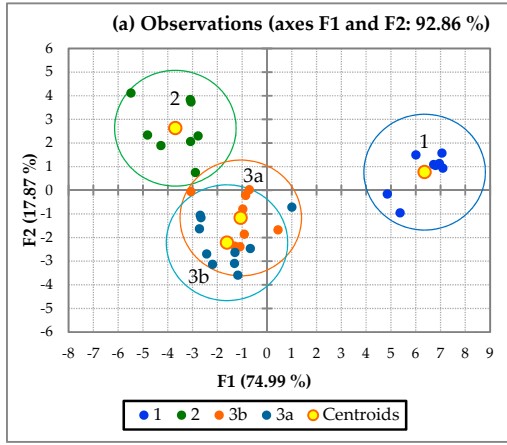 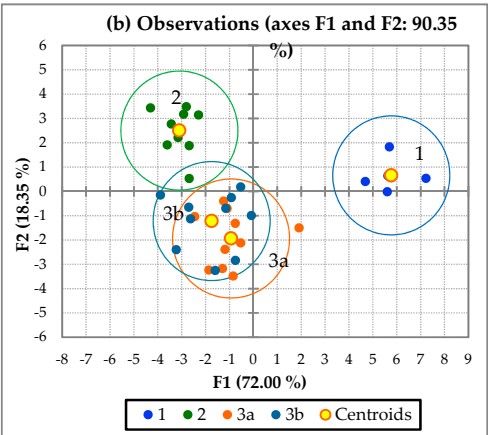

**Figure 5.** Results from DA among 4 groups in each season (1, 2, 3a, 3b): (**a**) dry season, (**b**) wet season. In both the dry and wet seasons, there was a large discriminant between cluster 1 and the other clusters, while cluster 3a and 3b had fewer discriminant features. In the dry season, F1 and F2 explained 74.99% and 17.87% of the total variation, respectively. In the wet season, F1 and F2 explained 72.01% and 18.35% of the total variation, respectively.

### 3.3. Data Structure Determination and Source Identification

In PCA, eigenvalues were used to identify principal components (PCs) that could be retained. An eigenvalue measures the significance of the factors. Scree plots were obtained based on the pronounced change of slope after the third eigenvalue, in which eigenvalues greater than or equal to 1 were considered [59,85,86]. In this study, the first four eigenvalues in both seasons were selected for further analysis. These PCs were identified with eigenvalues greater than 1, and explained 78.36% and 70.70% of total variation in the dry and wet seasons, respectively.

Figure 6 shows that PC1 explained 33.91% of the total variation in the dry season. This component positively and largely contributed to mineral/physical parameters (EC and turbidity), and inorganic parameters ($NO_2^-$ and $NO_3^-$); and negatively contributed to pH concentration. It was determined that 21.16% of PC2 was contributed by inorganic nutrient-related water quality parameters ($NO_2^-$ and $NO_3^-$), and pH. The concentrations of DO and TC were considered as less important since the loading coefficients (eigenvectors) were low in these two parameters in both seasons. These components revealed the importance of the effect of EC, Turb, and $PO_4^{3-}$ in the dry season. The high $NO_2^-$ and $NO_3^-$ values indicated that a high volume of fertilizers had been used, causing the spread of toxins which pose risk to aquatic life and human health. Wilbers et al. [87] indicated that the $NO_3^-$ concentrations were still slightly higher after treatment, while the other parameters were decreasing, including EC, $NH_4^+$, $NO_2^-$ concentrations in the surface water in the VMD. It was also found that most of these components appeared in cluster 1, in which most of the sampling sites were located in the secondary canals inside the full-dike protected area.

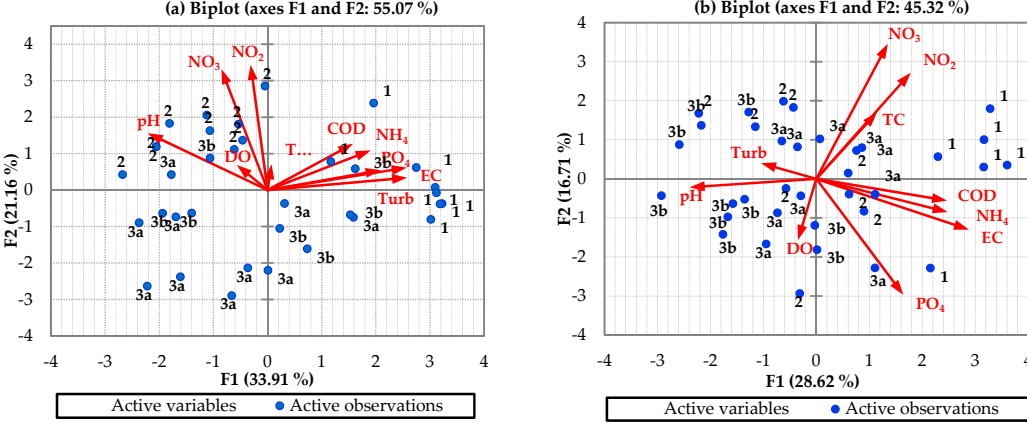

**Figure 6.** PC1 and PC2 loading of water parameters (**a**) in the dry season and (**b**) in the wet season. (**a**) shows a new coordinate axis in which F1 and F2 explained 33.91% and 21.16% of the total variance, respectively. The high variation of EC, Turb, pH, $NO_2^-$, $NO_3^-$ were found. Cluster 1 obtained high EC, Turb, COD, and $NH_4^+$. (**b**) shows a new coordinate axis in which F1 and F2 explained 28.62% and 16.71% of the total variance, respectively. The concentrations of EC, $NH_4^+$, COD, and $NO_3^-$ exhibited a high degree of variation in the wet season. Cluster 1 also had high EC, $NH_4^+$, and COD during the wet season. DO and TC exhibited low variation in both seasons.

In the wet season, PC1 explained 28.62% of the total variance, and most positively contributed to mineral (EC) and inorganic matters (COD, and $NH_4^+$), and negatively contributed to pH. This component revealed the importance of inorganic components over physically based water qualities, except for pH. PC2 in the wet season explained 16.71% of the total variance, and highly positively contributed to the inorganic nutrient-related water quality parameters ($NO_2^-$ and $NO_3^-$) and highly negatively contributed to $PO_4^{3-}$. These components demonstrated that DO, turbidity, and TC were less important in accounting for water quality variance. Furthermore, these components showed the highest EC and inorganic components (COD, $NH_4^+$, $NO_2^-$, and $NO_3^-$), which were found in cluster 1, where most of the sampling sites were located in the secondary canal inside the full-dike system.

As shown in Figure 6, the first component (PC1) and the second component (PC2) were highly influenced by most of the variables in both seasons. However, this result explains the difficulty in identifying which parameters are more important than others in influencing water quality variations in this study. Therefore, we conducted principal factor analysis (PFA) to circumvent the ambiguity in the data, and for the correlations between variables and factors for each season in this study area, we regarded an 85% correlation coefficient value to indicate the importance of the water quality parameters for seasonal variations at *p*-values of less than 0.05. Thus, EC, turbidity, $NO_2^-$, and $NO_3^-$ were identified as the most important parameters and contributed positively to water quality variations in the dry season. However, in the wet season, EC was identified as the most important parameter to water quality variations. Therefore, our findings reveal that EC is an important factor for water quality variations in the study area for both seasons.

### 3.4. Water Quality Assessment

Table 5 and Figure 7 show the values and display water quality index for the protection of aquatic life based on clusters-grades, $WQI_{al}$. In the dry season, most clusters were identified as "poor" water quality level; especially low-grade in cluster 1 were detected in both dry and wet seasons ($WQI_{al}$ = 3.7 and 4.7-grades, respectively). Although cluster 2 was found slightly low values of $NO_3^-$ and $PO_4^{3-}$ in the wet season, cluster 2 was considered as a "poor" water quality level in two seasons. The $WQI_{al}$ in cluster 3b was also considered as a "poor" level of water quality in dry and wet seasons even though the $QWI_{al}$ was close to "moderate" threshold in the wet season. Only $WQI_{al}$ in clusters 3a was approximately 7-grade in the wet season and cluster 3a was identified as a

"moderate" water quality level. Most sites in cluster 3a in the main river, and the main canal inside the semi-dike protected area. Cluster 3b, high turbidity concentration in wet was found in most sites in primary canals. Overall, the grades of WQI$_{al}$ were low mainly due to the extreme high EC, followed by NH$_3$, T-P, and NO$_{2-3}$-N. The high concentrations of NO$_{2-3}$-N and T-P may come from chemical fertilizers and pesticides, and domestic wastes. Moreover, the dike and sluice gate systems caused an increase in the concentrations when blocked water in the long-standing channels and the water drainage was controlled by the sluice gates operations calendar [88,89].

**Table 5.** Water Quality Index for aquatic life (WQIal) in An Giang.

| Clusters | Cluster 1 | Cluster 2 | Cluster 3a | Cluster 3b |
|---|---|---|---|---|
| Dry season | 3.7 | 4.2 | 5.4 | 4.8 |
| Wet season | 4.7 | 6.2 | 7.1 | 6.4 |

Note: 10–9 for high quality, 9.5–9 for good quality, 9–7 for moderate quality, and <7 for poor quality.

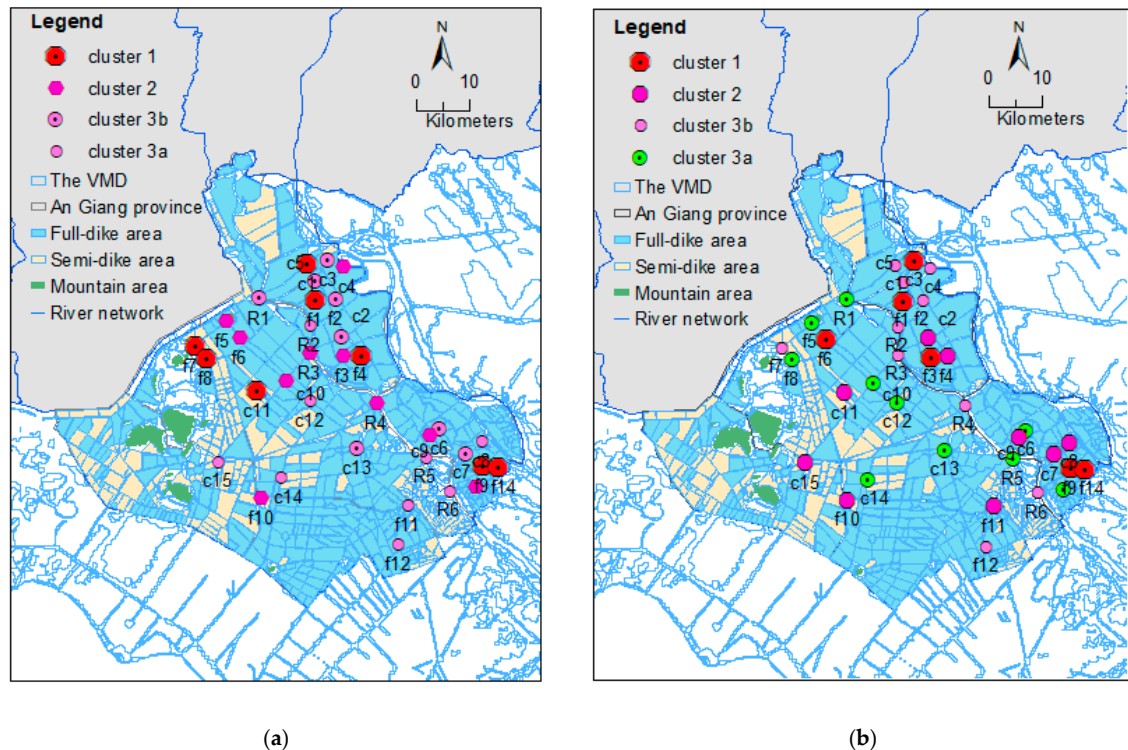

(**a**)                    (**b**)

**Figure 7.** Distribution of surface water quality in the dry season (**a**) and the wet season (**b**) obtained by cluster analysis. There are three main clusters (cluster 1, cluster 2, and cluster 3) and two sub-clusters (cluster 3a and 3b).

## 4. Discussion

Installations of dike systems for flood protection have a significant impact on hydrodynamics and water quality. Spatial and temporal variations in water quality inside and outside dike-protected areas were important for sustainable water resources management in the study area. We used multivariate statistical analysis to examine water quality parameters and their spatio-temporal variations during the dry and wet seasons in 2017. Water quality parameters showed clear seasonal variations in the study area. Water samples from inside the full-dike protected area were characterized by high pollution loads when compared to water samples from semi-dike systems and from outside the dike systems.

The EC was defined as the most important water quality parameter in both the wet and dry seasons in the study area. The EC was significantly correlated with dissolved solids (TDS), which might be due to the mixing of land-based pollutants entering into the water stream through runoff and leaching. Therefore, unlike other parameters, the EC in the dry season was comparatively higher than the EC during the wet season. This result was in agreement with the results in [90]. They showed that EC and TDS were important parameters. Moreover, [91] also found that EC and TDS necessarily influenced the concentrations of DO and BOD. However, DO concentration was not found to have a significant correlation with any other water quality parameters in either season in this study. It is possible that some other factors may have affected DO concentration, such as water temperature, photosynthetic activities by rice plants, TSS, BOD, and river discharge. Water level and river discharge may control the majority of the water quality parameters by the seasonal dilution of nutrient concentrations . Furthermore, [91] found that increased water level had a positive effect on water quality in their study, which was conducted in Lake Poyang, China. For example, water quality was best in summer, second best in autumn, and then in winter.

Due to the study area belongs to the Mekong River basin and thus, this study was partly affected by characteristics of inflow water and its quantity from the upper Mekong River basin of the trans-boundary river system [78,92]. In An Giang, the seasonal comparison of water quality parameters showed higher concentrations of $NO_3^-$, $PO_4^-$, and EC during the dry season than in the wet season (Figure 8). However, the mean concentrations and standard deviations of COD and $NH_4^+$ were higher in the wet season than in the dry season (Figure 8). This might be partly affected by other sources of pollution upstream of the Mekong River basin, due to the trans-boundary nature of the river systems [78,93]. For example, the mean COD concentrations in cluster 1 were 18 mgL$^{-1}$ (SD = 5.25 mg L$^{-1}$) and 39.67 mg L$^{-1}$ (SD = 13.37 mg L$^{-1}$) in the dry and wet seasons, respectively (please see Appendix A). The $NH_4^+$ concentrations were 1.23 mg L$^{-1}$ (SD = 0.5 mg L$^{-1}$) and 0.43 mg L$^{-1}$ (SD = 0.18 mg L$^{-1}$) in the dry and wet seasons, respectively. Chea et al. [74] determined that higher T-P and lower DO concentrations were often found in the wet season than in the dry season in the upper part of the Lower Mekong Delta basin. The authors explained that the low DO concentration in Cambodia was affected by the activities taking place in agricultural, domestic, and industrial zones in Cambodia in the period 1985–2010. Low concentrations of DO and high concentrations of T-P and $NH_4^+$ might affect aquatic life and human health, respectively [94]. Similar to in cluster 1, high concentrations of COD and $NH_4^+$ were found in the wet season in cluster 2 and cluster 3a,b.

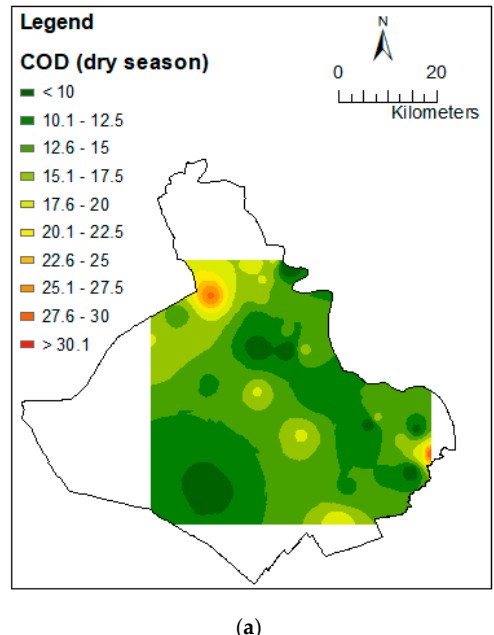

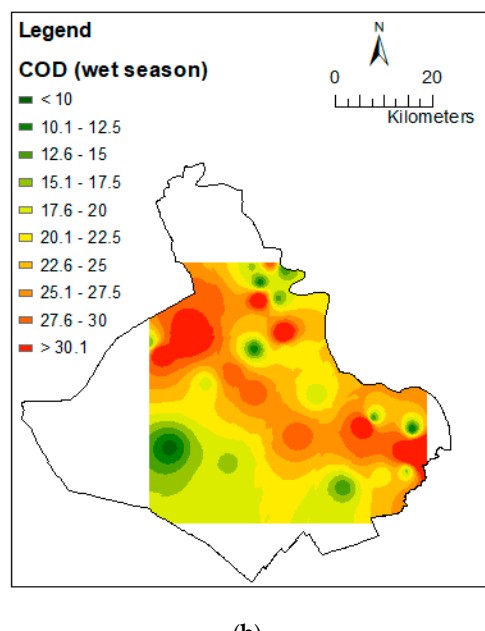

(a)　　　　　　　　　　　　　　　　　　　　　　　(b)

**Figure 8a.** Concentrations of COD displayed by IDW: (**a**) in the dry season, and (**b**) in the wet season, An Giang province, 2017.

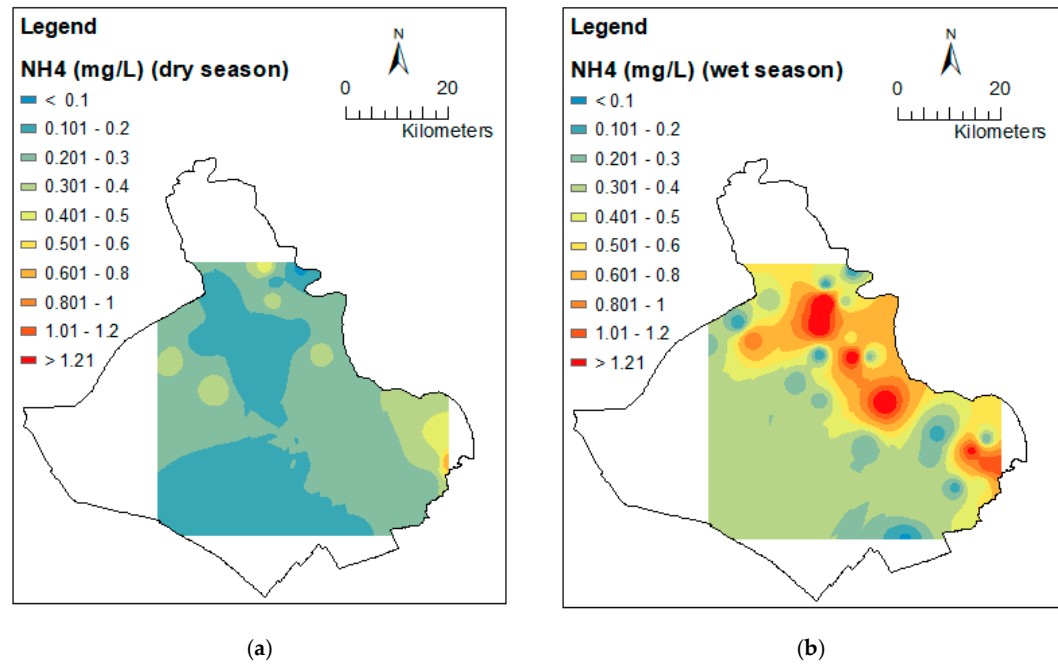

(**a**)                                                                                    (**b**)

**Figure 8b.** Concentrations of NH$_4^+$ displayed by IDW: (**a**) in the dry season, and (**b**) in the wet season, An Giang province, 2017.

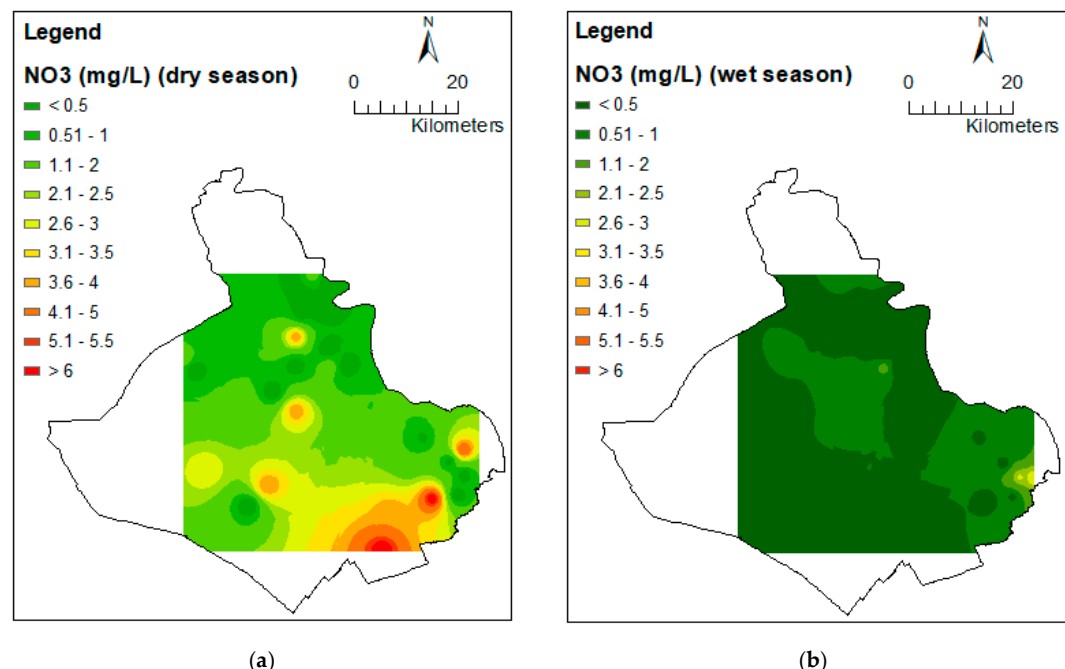

(**a**)                                                                                    (**b**)

**Figure 8c.** Concentrations of NO$_3^-$ displayed by IDW: (**a**) in the dry season, and (**b**) in the wet season, An Giang province, 2017.

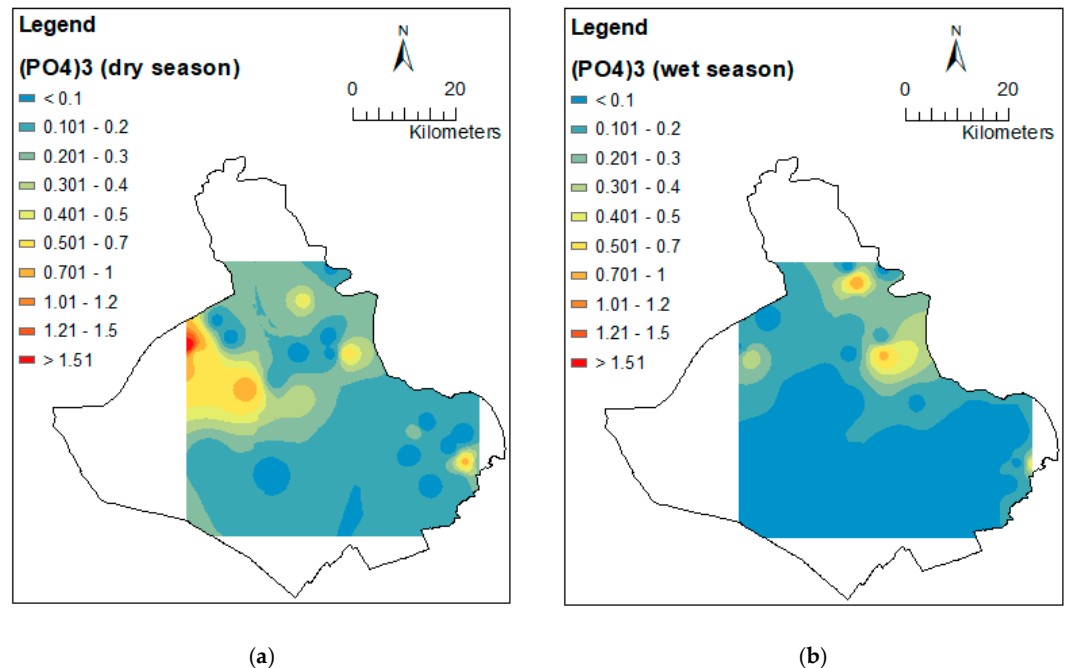

(**a**)　　　　　　　　　　　　　　　　　　(**b**)

**Figure 8d.** Concentrations of PO₄⁻ displayed by IDW: (**a**) in the dry season, and (**b**) in wet season, An Giang province, 2017.

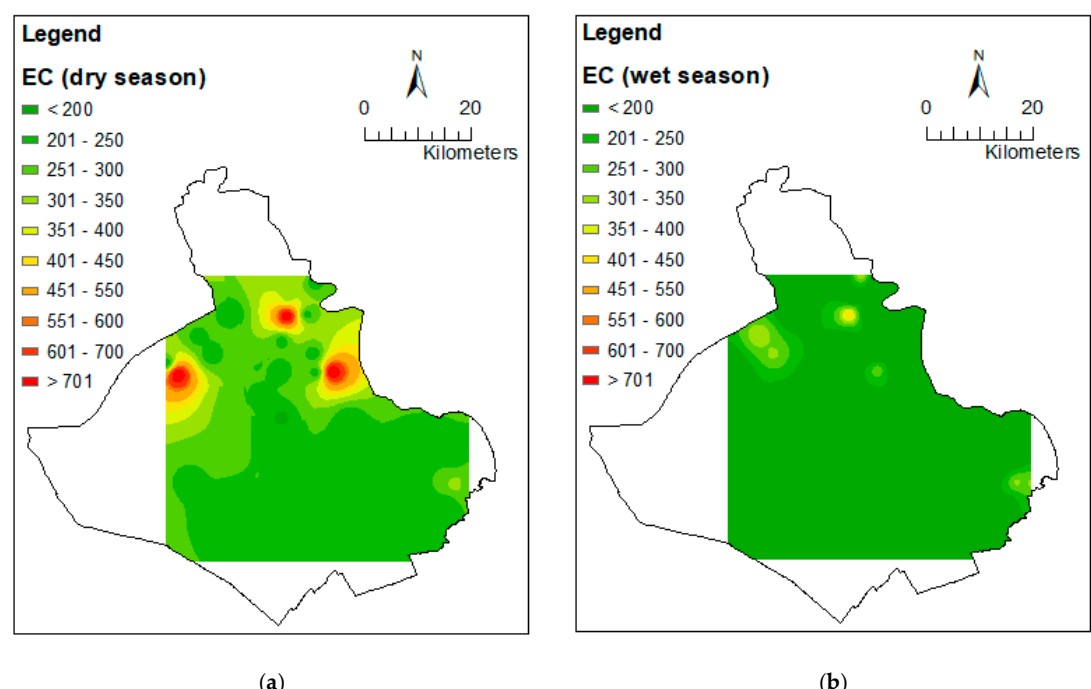

(**a**)　　　　　　　　　　　　　　　　　　(**b**)

**Figure 8e.** Concentrations of EC displayed by IDW: (**a**) in the dry season, and (**b**) in wet seasons An Giang province, 2017.

The DA presented different characteristics with regard to water quality parameters between the dry and wet seasons with CMs being 95.54% correct. HCA was successfully applied in clustering and indicated that cluster 1, located inside the dike-protected area, was different in terms of water quality characteristics from the other clusters. Specifically, PCA identified important parameters for

each season and for the whole year, with $p$ = 85%. However, the limitation of PCA is that its identification of the most important parameters is based only on the variance of parameter measurements. For example, the axes that exhibited high variance were considered to be principal components, while axes that exhibited low variance were treated as weak principal components.

Based on the $WQI_{al}$ results, "poor" surface water quality level was detected inside the full-dike system in An Giang province. Cluster 1, with sites located inside the full-dike area, was identified as being a water quality hot-spot. Cluster 1 had high concentrations of nutrients (N and P) in the dry season, and $NH_4^+$ and COD in the wet season. Furthermore, cluster 1 was identified as being the most significant discriminant among the clusters. The high nutrients in the dry season can be attributed to the wastewater that is discharged directly into the canal system from fishponds on secondary canals, as well as from households, markets, and tourists on primary canals. Moreover, people often live along the riverbanks of primary canals and rely on them for their livelihoods, while farmers often construct minor canals (or ditches) for irrigation and drainage from their rice fields [39]. Thus, the fresh water resources inside the dike system were not very suitable for aquatic life. The impact of dike systems on water quality could be associated with the operations calendar of sluice-gates [87,88]. Multi-dike systems are able to protect productivity; however, they have put pressure on soil and water quality within the region.

## 5. Conclusions

This study was conducted with aim of evaluating the seasonal variation of surface water quality samples in different dike-protected areas of the VMD. The multivariate analytical technique was successfully employed to distinguish water quality between the two seasons. EC was identified as the most important water quality parameter in both seasons. The main sources of pollution in the study area were combination of elements originating in local and upstream Mekong River resources. Therefore, the variation in surface water quality parameters during the wet season was higher than that during the dry season. The HCA method was used to divide the sample sites into 4 clusters. Cluster 1 (water samples from inside the full-dike system) showed "poor" water quality levels with reference to $WQI_{al}$ grades in both the dry and wet seasons.

The concentrations of water quality parameters were generally lower in the wet season than during the dry season because of high runoff causing the dilution of water quality parameters. An extremely high concentration of $PO_4^{3-}$ occurred during the dry season in the Northwest of An Giang, which is a tourist area with lots of human waste. Additionally, a high concentration of $NO_3^-$ was found inside the full-dike area located in the Thoai Son district, in the Southwest region of the province, during the dry season. This district contains the largest triple-rice crop area in An Giang.

Our study suggests that water quality in areas inside full dike-protected systems should be continuously monitored in both seasons. Furthermore, the continuous monitoring of some parameters, such as $PO_4^{3-}$ in Northeast An Giang and $NO_3^-$ in Southwest An Giang during the dry season, is recommended. With regard to the findings of high concentrations of $NH_4^+$ and COD in the Bassac River area during the wet season, we suggest monitoring water quality in the upper Mekong River basin. There is also a need to monitor water quality in the region using a real-time remote data acquisition system that is able to provide data in a timely fashion at a large scale. Dike protection systems have been useful in reducing flood hazards and supporting the intensification of rice cropping; however, it has had an adverse effect on the maintenance of water quality within the region. In recent years, the area devoted to triple-rice crops in An Giang has decreased, which might be beneficial in terms of restoring water and soil quality. It is expected that the government of An Giang will either adopt strategies to enhance water quality by reducing triple-rice crops or reconsider the construction and operation of full-dike protection areas.

**Author Contributions:** Conceptualization, M.K., and H.V.T.M.; methodology M.K., and H.V.T.M.; software, T.V.T., R.A., and H.V.T.M.; validation, M.K.; formal analysis, M.K., and H.V.T.M.; data curation, M.K., and H.V.T.M.; writing—original draft preparation, M.K., and H.V.T.M.; writing—review and editing, M.K., H.V.T.M., T.V.T., D.Q.T., K.N.L., R.A., M.M.R., and M.O.

**Funding:** This research received no external funding.

**Acknowledgement**: The authors thank the Vietnamese Ministry of Education and Training, Can Tho University, and Hokkaido University for supporting us to complete this research. This study is funded in part by the Can Tho University Improvement Project VN14-P6, supported by a Japanese ODA loan. Also, the authors appreciate the contributions made by the anonymous reviewers.

**Conflicts of Interest:** The authors declare no conflict of interest.

## Appendix A. Descriptive Statistics of the Collected Data

**Table A1.** Classification matrix for DA of seasonal change.

| From\to | Dry season | Wet season | Total | % Correct |
|---|---|---|---|---|
| **Standard mode** | | | | |
| Dry season | 33 | 1 | 34 | 97.14% |
| Wet season | 2 | 32 | 34 | 93.94% |
| **Total** | **35** | **33** | **68** | **95.54%** |
| **Backward stepwise mode** | | | | |
| Dry season | 31 | 3 | 34 | 91.43% |
| Wet season | 2 | 32 | 34 | 93.94% |
| **Total** | **33** | **35** | **68** | **92.68%** |
| **Forward stepwise mode** | | | | |
| Dry season | 31 | 3 | 34 | 91.43% |
| Wet season | 2 | 32 | 34 | 93.94% |
| **Total** | **33** | **35** | **68** | **92.68%** |

## Appendix B. Values of Water Quality Parameters in Dry Season (a) and Wet Season (b)

**Table A2.** Min, Max, Mean and Standard Deviation of water quality parameters at three clusters in the dry season.

| Variables | Cluster 1 | | | | Cluster 2 | | | | Cluster 3a | | | | Cluster 3b | | | |
|---|---|---|---|---|---|---|---|---|---|---|---|---|---|---|---|---|
| | Min | Max | Mean | SD | Min | Max | Mean | SD | Min | Max | Mean | SD | Min | Max | Mean | SD |
| DO | 4.98 | 7.83 | 5.93 | 0.93 | 5.11 | 7.96 | 6.67 | 1.06 | 4.98 | 7.55 | 5.66 | 0.78 | 5.66 | 8.24 | 7.38 | 0.78 |
| pH | 6.8 | 8 | 7.21 | 0.43 | 7.8 | 8.5 | 8.17 | 0.25 | 7.30 | 8.60 | 7.69 | 0.39 | 7.20 | 9.4 | 7.84 | 0.89 |
| EC | 220 | 810 | 486 | 270 | 190 | 250 | 224 | 18.10 | 200 | 330 | 228 | 37.06 | 200 | 320 | 245 | 48.70 |
| Turbidity | 30 | 56 | 40 | 9 | 3.75 | 30 | 14 | 9.63 | 3.64 | 46.72 | 14.38 | 12.13 | 3.34 | 28.31 | 13 | 8.66 |
| COD | 12 | 30 | 18 | 5.25 | 9 | 20 | 12.56 | 3.84 | 4.50 | 18 | 10.35 | 4.67 | 13 | 28 | 17.5 | 4.72 |
| $NH_4^+$ | 0.2 | 0.8 | 0.43 | 0.18 | 0.1 | 0.5 | 0.22 | 0.11 | 0.05 | 0.30 | 0.17 | 0.09 | 0.1 | 0.3 | 0.2 | 0.07 |
| $NO_2^-$ | 0.01 | 0.2 | 0.07 | 0.06 | 0.05 | 0.4 | 0.18 | 0.11 | 0.01 | 0.10 | 0.02 | 0.03 | 0.01 | 0.1 | 0.03 | 0.03 |
| $NO_3^-$ | 0.01 | 1.5 | 0.58 | 0.65 | 1.5 | 6 | 4.17 | 1.41 | 0.01 | 1.5 | 0.36 | 0.53 | 0.00 | 2.00 | 0.66 | 0.75 |
| $PO_4^{3-}$ | 0.2 | 1.8 | 0.7 | 0.51 | 0.02 | 0.4 | 0.13 | 0.13 | 0.01 | 0.80 | 0.14 | 0.24 | 0.02 | 0.30 | 0.13 | 0.10 |
| TC | 170 | 856 | 490 | 218 | 200 | 1040 | 573 | 233 | 144 | 856 | 587 | 195 | 300 | 1200 | 634 | 257 |

**Table A3.** Min, Max, Mean and Standard Deviation of water quality parameters at three clusters in the wet season.

| Variables | Cluster 1 | | | | Cluster 2 | | | | Cluster 3a | | | | Cluster 3b | | | |
|---|---|---|---|---|---|---|---|---|---|---|---|---|---|---|---|---|
| | Min | Max | Mean | SD | Min | Max | Mean | SD | Min | Max | Mean | SD | Min | Max | Mean | SD |
| DO | 6.59 | 7.25 | 6.92 | 0.25 | 6.60 | 7.42 | 7.04 | 0.25 | 6.85 | 9.00 | 7.75 | 0.78 | 6.85 | 7.24 | 7.04 | 0.14 |
| pH | 7 | 7.60 | 7.37 | 0.22 | 7.50 | 8.70 | 7.94 | 0.38 | 7.20 | 8.00 | 7.60 | 0.27 | 7.60 | 8.60 | 8.06 | 0.28 |
| EC | 280 | 430 | 329 | 52.35 | 70 | 160 | 116.6 | 30 | 110 | 350 | 158 | 71.15 | 100 | 140 | 117.7 | 13.94 |
| Turbidity | 13.5 | 80 | 38.8 | 23.80 | 4.34 | 45.38 | 21.14 | 14.26 | 22.40 | 48.20 | 33.47 | 9.31 | 33.8 | 68 | 45.89 | 11.44 |
| COD | 20 | 55 | 39.67 | 13.37 | 9 | 40 | 22.11 | 11.91 | 13 | 35 | 26.20 | 7.51 | 11 | 20 | 15.17 | 3.28 |
| $NH_4^+$ | 0.50 | 2 | 1.23 | 0.5 | 0.1 | 1.3 | 0.41 | 0.35 | 0.10 | 0.40 | 0.26 | 0.10 | 0.08 | 1.5 | 0.34 | 0.46 |
| $NO_2^-$ | 0.01 | 0.75 | 0.32 | 0.31 | 0.00 | 0.05 | 0.02 | 0.02 | 0.01 | 0.05 | 0.02 | 0.01 | 0.01 | 0.70 | 0.09 | 0.23 |
| $NO^-$ | 0.02 | 3 | 1.45 | 1.13 | 0.01 | 1 | 0.31 | 0.31 | 0.10 | 0.80 | 0.44 | 0.24 | 0.02 | 1 | 0.32 | 0.32 |
| $PO_4^{3-}$ | 0.07 | 0.80 | 0.35 | 0.26 | 0.02 | 0.50 | 0.10 | 0.15 | 0.03 | 0.40 | 0.10 | 0.11 | 0.02 | 1 | 0.18 | 0.31 |
| TC | 320 | 740 | 567 | 173 | 430 | 850 | 702 | 143 | 250 | 520 | 391 | 76 | 150 | 480 | 319 | 116 |

## Appendix C. Different Spatial Distribution of Water Quality between Dry and Wet Seasons

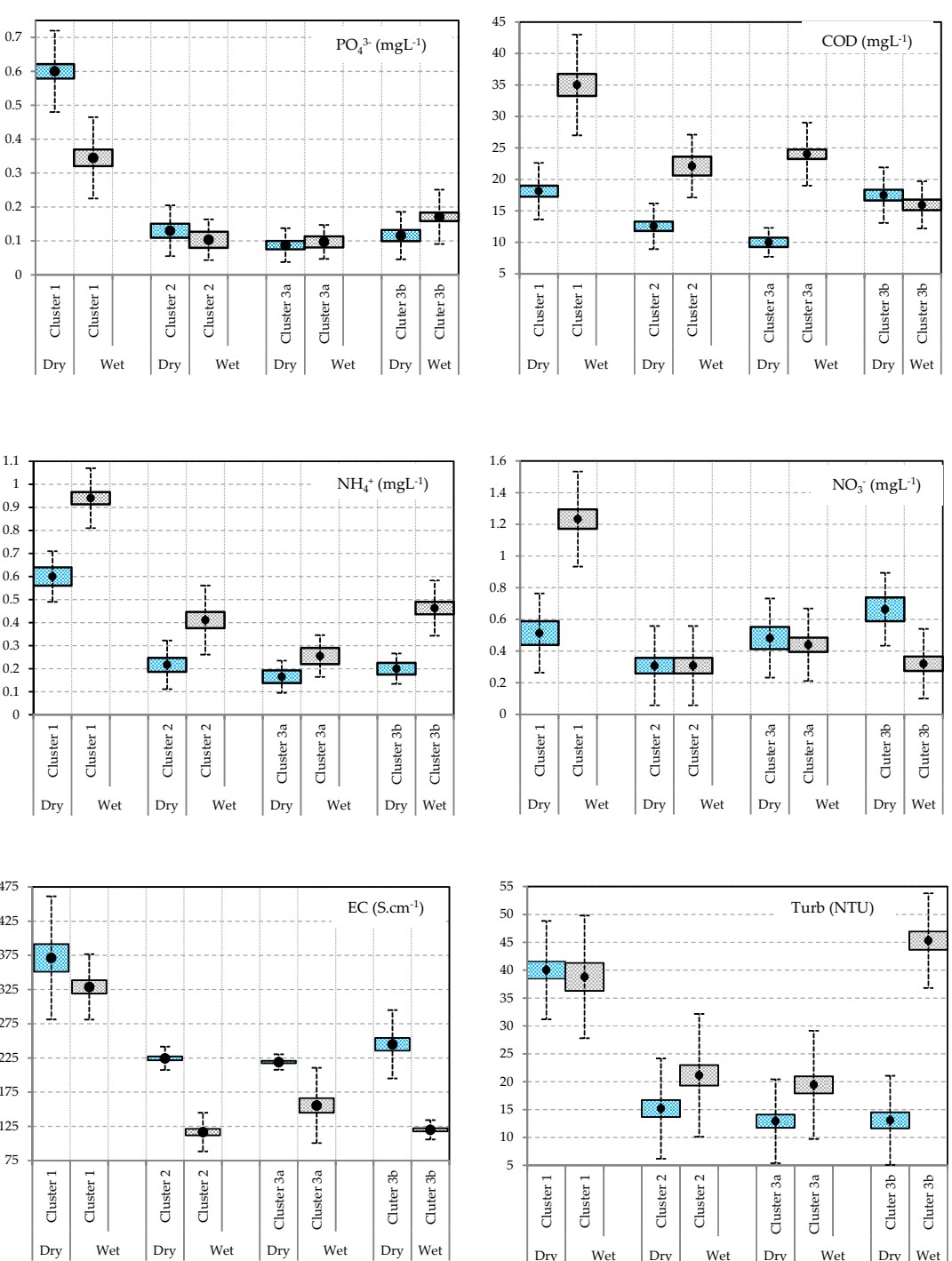

**Figure A1.** Different spatial distributions of water quality between dry and wet seasons.

## Appendix D. Correlations between Variables and Factors

**Table A4.** Correlations between variables and factors for each season.

| Parameters | Dry Season | | Wet Season | |
|:---:|:---:|:---:|:---:|:---:|
| | F1 | F2 | F1 | F2 |
| DO | −0.191 | 0.184 | −0.097 | −0.336 |
| pH | −0.746 | 0.415 | −0.670 | −0.046 |
| **EC** | **0.907 \*** | 0.164 | **0.907 \*** | −0.278 |
| **Turb** | **0.870 \*** | 0.091 | −0.293 | 0.086 |
| COD | 0.534 | 0.339 | 0.691 | −0.119 |
| $NH_4^+$ | 0.642 | 0.290 | 0.695 | −0.182 |
| $NO_2^-$ | −0.105 | **0.915 \*** | 0.499 | 0.582 |
| $NO_3^-$ | −0.287 | **0.884 \*** | 0.378 | 0.741 |
| $PO_4^{3-}$ | 0.705 | 0.145 | 0.460 | −0.635 |
| TC | 0.025 | 0.190 | 0.321 | 0.364 |

Note: * value is different from 0 at a significant level of 0.05.

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
