# Peer review of "Effects of Multi-Dike Protection Systems on Surface Water Quality in the Vietnamese Mekong Delta"

_water, doi:10.3390/w11051010_

Round 1

Reviewer 1 Report

line 41 chapter Introduction should be shortened

figure 1 the quality of figure is not sufficient; f,c not visible

line 217 table 2 didn't show the grade scale, text to be reconstructed

figure 7 the quality of figure is not sufficient

appendix A to be cancelled

From appendix figures and tables only that should stay which didn't multiplicate information from text tables and figures.

Author Response

Dear Reviewer #1,

The authors would like to express our sincere thank you for your helpful and insightful comments, which are very valuable in improving our manuscript. Please see our responses in bold text and in the revised manuscript (highlighted text) in the attached files. Thank you again.

Best regards,

Minh.

Reviewer 2 Report

The paper adopts an in depth statistical analysis of water quality data, but the results are not completely analysed and discussed.

The keywords should not contain the same words of the title

The introduction is not exhaustive on the literature review. The part of the study area should be moved and put under materials and methods, the study area.

Figure 8 in the discussion should be moved. They are still results.

All what is in the appendix should be part moved in the text and part deleted (table B1, C1, ..)

Conclusions are too general and short

Brackets for the charge of the ions are not needed

Please find below more detailed comments. I think the authors should work a lot on the text and then they can fix the sentences that are not correct, because some words are missing.

Line 34: it should be a point.

Line 72: move in materials and methods, the study area. Describe separately the hydraulic system and the cropping system

Figure 1: add a), b) ….

Lines 139:  “were measured” first samples are taken then measurements are done

Lines 150: I would delete this table

Line 168: the parameters in the equation (1) are not consistent with the description in 170-171

Line 210: conducted to be substituted with calculated

219: table 2 note are not clear

Line 223: add text before the table

Line 360-361: check the sentence

Line 387-390: check the sentence

Line 395. Problems in the sentence

Line 405: majority water quality ??

Author Response

Dear Reviewer #2,

The authors would like to express our sincere thank you for your helpful and insightful comments, which are very valuable in improving our manuscript. Please see our responses in bold text and in the revised manuscript (highlighted text) in the attached files. Thank you again.

Best regards,

Minh.

Reviewer 3 Report

The study of Minh et al. tackles an important problem of the water quality in the Vietnamese Mekong Delta in term of functioning of flood protection system, transformation of riverine ecosystem and intensification of agriculture. The authors tried to evaluate whether multi-dyke system in An Giang Province can affect the fluviogenius ecosystem and to identify the response of the surface/floodwater chemistry to changes of aquatic ecosystems. The authors compare the water conditions between the two seasons in the year 2017 (dry and wet).

    The manuscript brings important methodological data that can be useful for the water management. Also worthy of note is the comprehensive approach to the study design. In my opinion, the paper is well written, the topic is timely and will be of interest to the readers of the journal Water. I could not find any logical errors in the presentation and the approaches used. The following points may be considered while revising the article:

Specific Comments:

Introduction; line 75

The mean annual rainfall is about 1,400 mm accounting for 90% of the total annual rainfall.

This sentence is incomprehensible. Do you mean: The mean annual rainfall during monsoon seasons is about 1 400 mm…

Introduction; lines 88-89, Figure 1

Figure 1 is illegible.

- Geographical location of the study site - please add country names (Vietnam and mentioned in Discussion Cambodia) and change the blue color of the border of Mekong River Basin (it's confusing); please add the Mekong River, if possible;

- The border of An Giang Province is invisible;

- Please reorganize legend items – start with an area feature type (An Giang Province, Full-dyke, Semi-dyke, Mountain area), next line feature types (Rivers) and finally point feature types (Water sampling sites).

Results; line 223

Please do not start a chapter of the table.

Discussion; lines 441-447 and line 338

Please organize the numbering of the figures (Fig. 8-1 and Fig. 8 - tree times).

Please remove from the legends the subtitles: <Value>; please add unites to parameters.

Discussion; line 444; Fig. 8 left panel

< 30 replace with < 0.30

Author Response

Dear Reviewer #3,

The authors would like to express our sincere thank you for your helpful and insightful comments, which are very valuable in improving our manuscript. Please see our responses in bold text and in the revised manuscript (highlighted text) in the attached files. Thank you again.

Best regards,

Minh.

Round 2

Reviewer 2 Report

The paper has been improved a lot.